# Altered Plasma Fatty Acid Abundance Is Associated with Cachexia in Treatment-Naïve Pancreatic Cancer

**DOI:** 10.3390/cells11050910

**Published:** 2022-03-07

**Authors:** Kristyn Gumpper-Fedus, Phil A. Hart, Martha A. Belury, Olivia Crowe, Rachel M. Cole, Valentina Pita Grisanti, Niharika Badi, Sophia Liva, Alice Hinton, Christopher Coss, Mitchell L. Ramsey, Anne Noonan, Darwin L. Conwell, Zobeida Cruz-Monserrate

**Affiliations:** 1Division of Gastroenterology, Hepatology, and Nutrition, Department of Internal Medicine, The Ohio State University Wexner Medical Center, Columbus, OH 43210, USA; kristyn.gumpper@osumc.edu (K.G.-F.); philip.hart@osumc.edu (P.A.H.); valentina.pitagrisanti@osumc.edu (V.P.G.); b.niharika.reddy1808@gmail.com (N.B.); mitchell.ramsey@osumc.edu (M.L.R.); darwin.conwell@osumc.edu (D.L.C.); 2The James Comprehensive Cancer Center, The Ohio State University Wexner Medical Center, Columbus, OH 43210, USA; sophliva15@gmail.com (S.L.); coss.16@osu.edu (C.C.); anne.noonan@osumc.edu (A.N.); 3Program of Human Nutrition, College of Education and Human Ecology, The Ohio State University, Columbus, OH 43210, USA; belury.1@osu.edu (M.A.B.); cole.311@osu.edu (R.M.C.); 4College of Medicine, The Ohio State University, Columbus, OH 43210, USA; olivia.crowe@cchmc.org; 5Division of Pharmaceutics and Pharmacology, College of Pharmacy, The Ohio State University, Columbus, OH 43210, USA; 6Division of Biostatistics, College of Public Heath, The Ohio State University Wexner Medical Center, Columbus, OH 43210, USA; alice.hinton@osumc.edu; 7Division of Medical Oncology, Department of Internal Medicine, The Ohio State University Wexner Medical Center, Columbus, OH 43210, USA

**Keywords:** pancreatic cancer, fatty acids, sarcopenia, diabetes, hemoglobin, albumin, oleic acid, linoleic acid

## Abstract

Cachexia occurs in up to 80% of pancreatic ductal adenocarcinoma (PDAC) patients and is characterized by unintentional weight loss and tissue wasting. To understand the metabolic changes that occur in PDAC-associated cachexia, we compared the abundance of plasma fatty acids (FAs), measured by gas chromatography, of subjects with treatment-naïve metastatic PDAC with or without cachexia, defined as a loss of > 2% weight and evidence of sarcopenia (n = 43). The abundance of saturated, monounsaturated, and polyunsaturated FAs was not different between subjects with cachexia and those without. Oleic acid was significantly higher in subjects with cachexia (*p =* 0.0007) and diabetes (*p* = 0.015). Lauric (r = 0.592, *p* = 0.0096) and eicosapentaenoic (r = 0.564, *p* = 0.015) acids were positively correlated with age in cachexia patients. Subjects with diabetes (*p* = 0.021) or both diabetes and cachexia (*p* = 0.092) had low palmitic:oleic acid ratios. Linoleic acid was lower in subjects with diabetes (*p* = 0.018) and correlated with hemoglobin (r = 0.519, *p* = 0.033) and albumin (r = 0.577, *p* = 0.015) in subjects with cachexia. Oleic or linoleic acid may be useful treatment targets or biomarkers of cachexia in patients with metastatic PDAC, particularly those with diabetes.

## 1. Introduction

Pancreatic ductal adenocarcinoma (PDAC) is one of the deadliest cancers with a dismal 5-year survival rate of less than 10% [1]. Cachexia, a multi-systemic, cancer-related metabolic disorder that includes weight loss (either ≥ 5% weight loss, a BMI < 20, or ≥ 2% weight loss with sarcopenia), tissue wasting, and loss of appetite, occurs in up to 80% of PDAC patients, being more common as cancer stage progresses [2,3,4,5]. Cachexia is associated with shorter survival, reduced efficacy of cancer therapy, dysregulated metabolism, impaired physical function, and reduced quality of life in PDAC patients.

The metabolic changes that occur in patients with cachexia stem from accelerated breakdown of adipose and muscle tissue, which leads to weight loss, elevated plasma lipoproteins, and release of circulating fatty acids (FAs) [6]. The release of FAs during tissue wasting can alter the normal plasma concentration of saturated (SFAs), monounsaturated (MUFAs), and polyunsaturated FAs (PUFAs), as described for other pancreatic inflammatory diseases [7]. Changes in FA abundance are associated with several metabolic disorders, including diabetes and metabolic syndrome [8,9]. FA release has also been correlated with changes in the inflammation status of patients with cachexia, which was defined by greater than 5% or 10% weight loss [10,11]. To counteract cachexia-induced weight loss, patients are often prescribed appetite stimulants, anti-inflammatory medications, nutritional supplements, and exercise [12,13]. Fish oils high in omega-3 PUFAs are often recommended as nutritional supplements and are reported to mitigate cachexia symptoms by reducing inflammation signaling, which improves cellular metabolism and increases muscle and fat mass. However, any improvements are mild, and compliance in clinical trials is low [14,15,16,17,18,19]. The limited effectiveness of current FA supplementation therapies may also be partially due to targeting FAs that are not dysregulated in all cachexia patient populations. Therefore, defining the changes in FA abundance that occur in cachexia patients is critical to the development of effective nutritional interventions.

The goal of this study was to identify changes in plasma FA abundance in metastatic PDAC associated with sarcopenic cachexia to better understand the metabolic changes that occur in cachexia and to discover new targets for cachexia treatments. We also evaluated the relationship between FA abundance and other clinical factors that may contribute to sarcopenia, including age, body mass index (BMI), diabetes status, and levels of hemoglobin and albumin.

## 2. Materials and Methods

### 2.1. Study Population

We performed an auxiliary study using baseline plasma samples from a cohort of treatment-naïve metastatic PDAC subjects enrolled in a phase 2, multicenter, randomized, and controlled study (NCT01280058) that occurred between February 2011 and February 2014 [20]. Subjects were selected for the study presented here based on the availability of clinical data, computed tomography (CT) scans, and plasma from the baseline blood draw of the clinical trial, resulting in a final cohort of 43 subjects (Figure 1). The original trial and current study were approved by The Ohio State University Wexner Medical Center Institutional Review Board (2010C0055) and followed the protocols and standards of the Declaration of Helsinki. Clinical data from study subjects were abstracted using a standardized form, including age at enrollment, sex, height, weight, CT images of the third lumbar vertebra, diabetes status, hemoglobin, and albumin. 

### 2.2. Cachexia Classification

In this study, cachexia was determined using the definition published by Fearon et al. in 2011 [21]. We stratified subjects using a >2% weight loss cutoff and CT evidence of sarcopenia. CT scans were evaluated for skeletal muscle area at the third lumbar vertebra (L3) [22]. Images were analyzed for cross-sectional area (cm^2^) using the Slice-O-Matic software V4.3 (Tomovision, Montreal, Quebec, QC, Canada), as previously described [23,24]. For normalization, the skeletal muscle index (SMI) (cm^2^/m^2^) was calculated by dividing the cross-sectional area of the skeletal muscle by the height of the patient squared (m^2^). Sarcopenia was defined using the SMI stratified by sex and BMI. Subjects were classified with sarcopenia if they met the following criteria: SMI < 41 cm^2^/m^2^ for females with any BMI, < 43 cm^2^/m^2^ for males with a BMI < 24.9 kg/m^2^, and <53 cm^2^/m^2^ for males with a BMI > 25 kg/m^2^ [25]. Subjects with weight loss <2% or no CT evidence of sarcopenia were classified as subjects without cachexia (no cachexia), and subjects with both >2% weight loss and sarcopenia were classified as subjects with cachexia (cachexia).

### 2.3. Blood Collection and Processing

Random (i.e., non-fasting) blood collections were performed in all subjects at the baseline study visit in a tube containing sodium heparin as an anti-coagulant. Aliquots of plasma were separated and frozen in a −80 °C freezer until samples were batch analyzed for FA abundance.

### 2.4. Fatty Acid Analysis

Total lipids were extracted from plasma samples using the 2:1 chloroform:methanol method and then washed with 0.88% potassium chloride, as previously described [26]. FA methyl esters were then prepared using the 5% hydrochloric acid in methanol method [27], heated overnight at 76 °C, and extracted with hexane. FA methyl esters were analyzed by gas chromatography using a 30-m Omegawax 320 fused silica capillary column (Supelco Bellefonte, PA, USA). The oven temperature was set to 175 °C, heated at a rate of 3 °C/min until reaching 220 °C, and cooled at a rate of 5 °C/min until reaching 210 °C. The flow rate of carrier gas helium was set to 30 mL/min. FA measurements were recorded as a percentage of the total FAs identified (% area) [28] and compared to FA methyl ester standards (Matreya LLC, Pleasant Gap, PA, USA; Supelco, Bellefonte, PA, USA; and Nu-Check Prep Inc, Elysian, MN, USA). We assessed individual FAs as well as FAs grouped by saturation type abundance: SFA, MUFA, or PUFA.

### 2.5. Statistical Analysis

FA abundance was compared using an uncorrected Fishers Least Significant Difference (LSD) test. Saturation type, BMI, and the presence or absence of diabetes as indicated in the clinical chart was assessed using two-way ANOVA with Sidak’s multiple comparisons for pairwise differences. Differences in categorical variables were analyzed with chi-square tests. Pearson correlations of each FA with continuous variables, such as age, hemoglobin, albumin, or BMI, were performed. Positive R-values indicate positive correlations. Pearson correlations were used to determine the probability that the correlation observed occurred due to random sampling when p was equal to or less than 0.05. Data analyses were performed in Prism (GraphPad, version 9.0, San Diego, CA, USA).

## 3. Results

### 3.1. Characteristics of the Study Population

Of the 43 subjects in this study (Figure 1), 41.9% (n = 18) had cachexia and 58.1% (n = 25) did not. Age, sex, BMI, diabetes, hemoglobin, and albumin were not significantly different between the subjects with and without cachexia (Table 1). As expected, the percent weight loss and SMI were significantly different, as these were used for cachexia classification.

### 3.2. Oleic Acid Abundance Is Elevated in Subjects with Cachexia

We then investigated differences in plasma FA abundance between subjects with and without cachexia. There was no difference in overall FA abundance based on saturation type (Figure 2A). Oleic acid was the only MUFA that demonstrated higher abundance in subjects with cachexia (*p =* 0.0007) (Figure 2B). No individual SFAs or PUFAs exhibited changes in abundance between subjects with and without cachexia (Figure 2C,D).

### 3.3. Lauric and Eicosapentaenoic Acid Abundance Correlate with Age in Subjects with Cachexia

Aging impacts energy metabolism and modulates FA abundance in adipose tissue [29,30]. As PDAC is more common in older populations [31], we assessed whether plasma FA abundance was related to age and cachexia. We found that lauric (r = 0.592, *p =* 0.0096) and eicosapentaenoic (r = 0.564, *p =* 0.015) acids were positively correlated with age in cachexia patients, suggesting that the abundance of these FAs increases with age (Table 2).

### 3.4. BMI and Diabetes Alter Oleic and Linoleic Acid Levels in Subjects with Cachexia

Obesity is a strong risk factor for PDAC and is associated with changes in free FAs related to insulin resistance [32,33,34,35]. Additionally, sarcopenic obesity, where obesity occurs simultaneous with muscle wasting, occurs in up to 15% of patients with respiratory or gastrointestinal cancers [36]. Therefore, we assessed whether any FAs correlated with BMI. As BMI increased in subjects without cachexia, docosapentaenoic omega-6 (n6) acid abundance decreased (r = –0.550, *p =* 0.004) (Table 3). In subjects with cachexia, docosapentaenoic n6 acid abundance was no longer correlated with BMI. In contrast, vaccenic acid abundance increased as BMI increased in subjects with cachexia (r = 0.483, *p =* 0.042).

We also explored whether FA abundance was associated with diabetes status. Oleic acid was significantly higher in subjects with diabetes (*p =* 0.015) (Figure 3A). Additionally, linoleic acid was reduced overall in subjects with diabetes (*p =* 0.018) (Figure 3B). Gondoic acid was higher overall in subjects with diabetes compared to those without diabetes (*p =* 0.047) (Figure 3C). Similarly, α-linolenic acid was elevated overall in subjects with diabetes (*p =* 0.046) (Figure 3D). A low ratio of palmitic to oleic acid can indicate the presence of diabetes [37], which we confirmed in subjects with diabetes (*p =* 0.021) in our study. Interestingly, we observed a trend toward lower palmitic to oleic acid ratios in subjects with cachexia compared to those without cachexia (*p =* 0.092) (Figure 3E).

### 3.5. Some FAs Correlate with Other Indicators of Cachexia

Levels of hemoglobin and albumin are typically lower in patients with cachexia, and these clinical factors are used in some definitions of cachexia [38,39]. Additionally, albumin is an FA transporter in blood [38,39,40]. Although hemoglobin and albumin were not significantly lower in patients with cachexia (Table 1), we still assessed whether any FAs correlated with either clinical value. In subjects without cachexia, lauric acid negatively correlated (r = –0.423, *p =* 0.035) and gondoic acid positively correlated (r = 0.406, *p =* 0.044) with hemoglobin levels. In subjects with cachexia, palmitic (r = −0.489, *p =* 0.046) and linoleic (r = 0.519, *p =* 0.033) acids positively correlated with hemoglobin levels (Table 4). When comparing FA abundance with albumin (Table 5), lauric and vaccenic acids were strongly negatively correlated with albumin in subjects without cachexia (r = –0.533, *p =* 0.006 and r = –0.560, *p =* 0.004; respectively). Additionally, γ-linolenic acid was positively correlated with albumin in subjects without cachexia. In subjects with cachexia, vaccenic remained strongly negatively correlated with albumin (r= −0.646, *p =* 0.005), whereas linoleic was positively correlated with albumin (r = 0.577, *p =* 0.015).

## 4. Discussion

Although cachexia is prevalent among metastatic PDAC patients and worsens patient outcomes, the nutritional therapies used for cachexia are not highly effective at mitigating symptoms. Therefore, we conducted a small retrospective study to identify changes in plasma FA abundance in metastatic PDAC associated with sarcopenic cachexia to better understand the metabolic changes in cachexia and to discover new targets for cachexia treatments.

The omega-9 MUFA oleic acid was significantly elevated in subjects with cachexia compared to those without (Figure 2). This correlation is consistent with previous studies using alternative definitions of cachexia, such as ≥ 5% weight loss [10]. Additionally, we found that oleic acid was higher in patients with diabetes (Figure 3). Oleic acid contributes to insulin sensitivity, and its accumulation in muscle is lower in subjects with type 2 diabetes compared to those without [41,42]. Similar to metformin, which is commonly used to treat type 2 diabetes, oleic acid blocks the palmitic acid-mediated inhibition of adenosine monophosphate-activated protein kinase activity. If oleic acid uptake into metabolically active tissues, such as muscle, is inhibited in the context of a pancreatic disorder, such as diabetes, a similar phenomenon may occur in PDAC-associated cachexia, leading to elevated oleic acid levels in the plasma. Consistent with this hypothesis, we observed significantly lower palmitic:oleic acid ratios in subjects with cachexia and a trend for lower ratios in subjects with both cachexia and diabetes (Figure 3). The higher plasma levels of oleic acid in diabetic subjects and cachexia diabetic subjects in this cohort may result from the body compensating for malnutrition caused by PDAC or loss of oleic acid uptake into tissue, causing an accumulation in the plasma. A follow-up study on oleic acid metabolism or supplementation in malnutrition would elucidate this mechanism.

Omega-3 PUFA supplementation has been suggested for the treatment of cancer-related cachexia symptoms [14,16,43]. The absorption and metabolism of omega-3 PUFAs found in fish oils, such as eicosapentaenoic acid or docosahexaenoic acid, are theorized to improve cachexia by inhibiting proteolysis, lipolysis, and nuclear factor kappa B-induced inflammation [14,16,44]. However, some clinical trials indicate these FAs may not be as effective as once thought [45,46]. Indeed, we did not observe any differences in the abundance of PUFAs between subjects with cachexia compared to those without cachexia. However, we did find a strong positive correlation between age and eicosapentaenoic acid in subjects with cachexia only (Table 2). Previous studies have reported age-related increases in eicosapentaenoic and docosahexaenoic acids in the plasma fractions and red blood cells of “healthy” individuals [47,48]. Our results suggest that cachexia may intensify or accelerate age-related elevation of eicosapentaenoic acid, which warrants further investigation. The similarity of omega-3 PUFA abundance in our cohort provides some explanation for the ineffectiveness of omega-3 PUFA supplementation in metastatic PDAC subjects with cachexia.

Linoleic acid, an omega-6 PUFA, was significantly lower in subjects with diabetes and positively correlated with both albumin and hemoglobin in subjects with cachexia only (Figure 3, Table 4 and Table 5). High linoleic acid levels are associated with reduced risk in type 2 diabetes mellitus, improved insulin sensitivity, and reduced inflammation [49]. In addition to improving glycemic control, linoleic acid supplementation in women with type 2 diabetes mellitus increases lean mass and reduces trunk fat mass compared to conjugated linoleic acid supplementation [50], suggesting a potential benefit in diabetic PDAC patients similar to the present cohort. Mitochondrial dysfunction also contributes to insulin resistance and type 2 diabetes mellitus [41,51]. Linoleic acid improves the fatty acid composition of mitochondrial cardiolipins by increasing the levels of tetralinoleoyl (L_4_) cardiolipins, which are beneficial to mitochondrial function [52]. Although hemoglobin and albumin are reportedly lower in subjects with cachexia, we did not observe any differences in our cohort between subjects with cachexia and those without. However, we did observe a positive correlation between linoleic acid and these factors. Therefore, increasing dietary linoleic acid may provide nutritional support for patients experiencing cachexia.

This study utilized a unique treatment-naïve metastatic PDAC patient population to identify changes in FA abundance, albeit with a few limitations. Due to the retrospective study design, we were unable to completely characterize all aspects of cachexia, such as muscle function and appetite [38]. Similarly, FA analyses were performed using available non-fasted plasma samples instead of red blood cells, which are resistant to day-to-day changes in FA abundance caused by diet [53]. Additionally, we could not assess the impact of exocrine pancreatic function, physical activity, or diet on FA composition due to lack of survey data. Future studies of cachexia in larger PDAC patient populations with and without metastasis should be designed to collect this information to better inform the overall metabolic status of each individual and better track these metabolic changes across the stages of disease.

Despite the limitations of this retrospective study, we found differences in FA abundance that were associated with sarcopenic cachexia in treatment-naïve metastatic PDAC. A more extensive prospective study that addresses diet and exocrine function is necessary to confirm the FA abundance patterns identified here and to determine whether FAs are viable as novel therapies for cancer-associated sarcopenic cachexia.

## Figures and Tables

**Figure 1 cells-11-00910-f001:**
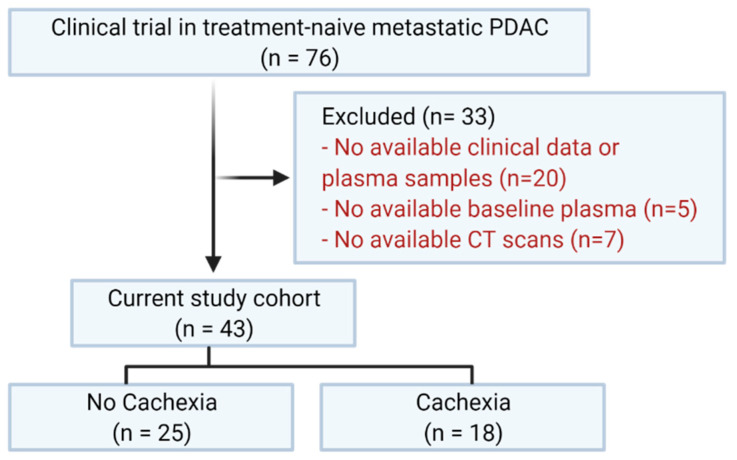
Subject selection flow chart. Created using Biorender.com.

**Figure 2 cells-11-00910-f002:**
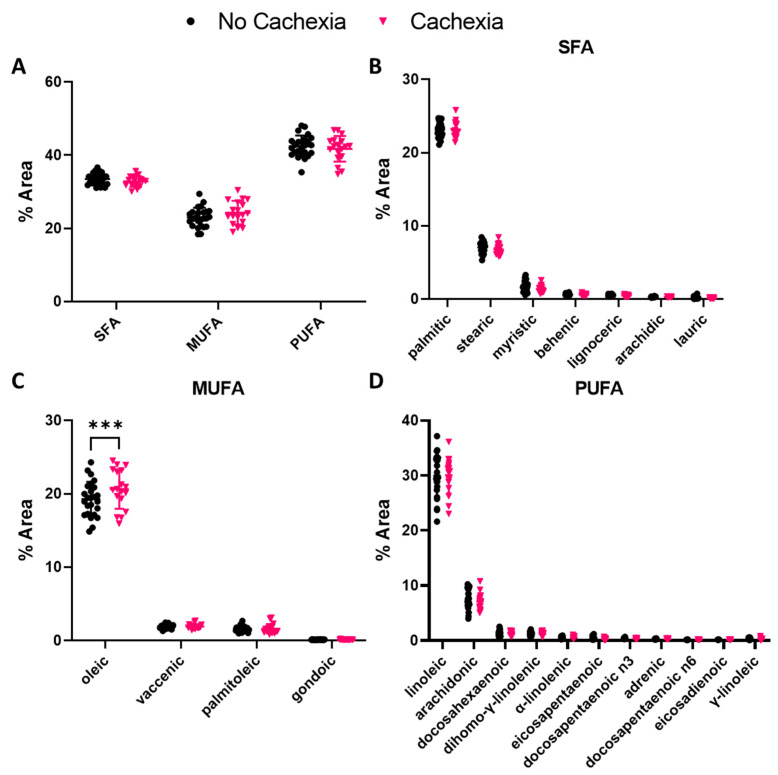
Fatty acid (FA) abundance in pancreatic ductal adenocarcinoma subjects based on cachexia. (**A**) FA abundance presented as % area and grouped by FA saturation type. Two-way ANOVA used to assess significance. (**B**) Individual SFA, (**C**) MUFAs, or (**D**) PUFAs assessed by two-way ANOVAs with multiple comparisons and no corrections. Data are presented as mean ± SD. *** *p ≤* 0.001.

**Figure 3 cells-11-00910-f003:**
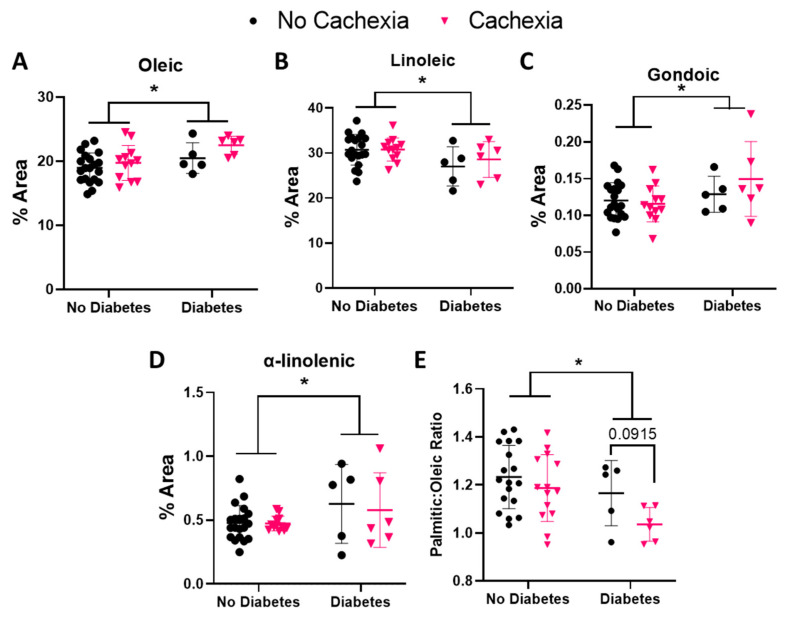
Fatty acid (FA) abundance relative to diabetes status in subjects with pancreatic ductal adenocarcinoma and cachexia. Percent area of (**A**) oleic, (**B**) linoleic, (**C**) gondoic, (**D**) α-linolenic acids were compared by cachexia and diabetes using a two-way ANOVA with Sidak’s correction for multiple comparison. (**E**) Ratio of the % area of palmitic acid to oleic acid using a t-test. Data are presented as mean ± SD. FAs were selected based on significant differences. * *p ≤* 0.05.

**Table 1 cells-11-00910-t001:** Baseline characteristics of the study subjects.

	No Cachexia (n = 25)	Cachexia (n = 18)	*p*-Value
Mean age (years)	62.9 ± 11.1	60.5 ± 10.5	0.480
Male sex, n (%)	18 (72.0)	10 (55.6)	0.338
% Weight loss	4.3 ± 7.5	11.1 ± 8.5	0.002 *
Skeletal muscle index (SMI)	49.6 ± 9.3	40.9 ± 5.9	0.0002 *
Body mass index (BMI) (kg/m^2^) 18.5–24.9 (% Normal) 25–29.9 (% Overweight) ≥ 30 (% Obese)	7 (28.0) 11 (44.0) 7 (28.0)	5 (27.8) 8 (44.4) 5 (27.8)	0.999
Diabetes present, n (%)	5 (20)	6 (33.3)	0.481
Hemoglobin (g/dL)	13.5 ± 1.8	13.5 ± 1.1	0.953
Albumin (g/dL)	3.9 ± 0.6	3.9 ± 0.4	0.652

Starred *p*-values indicate statistical significance < 0.05 by Student’s *t*-test.

**Table 2 cells-11-00910-t002:** Correlation between FA abundance and age stratified by cachexia status.

Fatty Acid	No Cachexia R	Cachexia R
Lauric	0.364	0.592 **
Myristic	0.032	0.307
Palmitic	0.281	−0.030
Palmitoleic	−0.300	−0.115
Stearic	−0.071	0.381
Oleic	−0.187	−0.083
Vaccenic	−0.037	−0.204
Linoleic	−0.048	−0.306
γ-Linolenic	−0.055	0.271
α-Linolenic	−0.002	0.137
Arachidic	0.116	0.107
Gondoic	0.029	−0.169
Eicosadienoic	−0.170	−0.058
Dihomo-γ-Linolenic	−0.109	0.213
Arachidonic	0.087	0.349
Eicosapentaenoic	0.364	0.564 *
Behenic	−0.086	0.094
Adrenic	−0.053	0.151
Docosapentaenoic n6	0.059	0.130
Docosapentaneoic n3	0.323	0.378
Lignoceric	0.114	0.083
Docosahexaneoic	0.355	0.318

Starred numbers indicate statistical significance. * *p <* 0.05, ** *p <* 0.01.

**Table 3 cells-11-00910-t003:** Correlation between FA abundance and BMI stratified by cachexia status.

Fatty Acid	No Cachexia R	Cachexia R
Lauric	−0.250	−0.287
Myristic	0.199	−0.281
Palmitic	−0.045	0.162
Palmitoleic	−0.99	0.291
Stearic	−0.099	−0.101
Oleic	0.070	0.426
Vaccenic	−0.014	0.483 *
Linoleic	0.116	−0.258
γ-Linolenic	−0.166	−0.293
α-Linolenic	0.386	−0.052
Arachidic	−0.187	0.094
Gondoic	−0.026	0.304
Eicosadienoic	0.013	−0.207
Dihomo-γ-Linolenic	−0.255	−0.204
Arachidonic	−0.152	−0.169
Eicosapentaenoic	0.009	−0.420
Behenic	−0.301	−0.201
Adrenic	−0.37	−0.067
Docosapentaenoic n6	−0.550 *	−0.135
Docosapentaneoic n3	−0.342	−0.334
Lignoceric	−0.258	−0.433
Docosahexaneoic	0.074	−0.111

Bolded numbers indicate statistical significance. * *p <* 0.05.

**Table 4 cells-11-00910-t004:** Correlation between FA abundance and hemoglobin stratified by cachexia status.

Fatty Acid	No Cachexia R	Cachexia R
Lauric	−0.423 *	0.143
Myristic	−0.274	−0.123
Palmitic	−0.187	−0.489 *
Palmitoleic	−0.270	−0.102
Stearic	−0.089	0.003
Oleic	−0.004	−0.419
Vaccenic	−0.130	0.048
Linoleic	0.232	0.519 *
γ-Linolenic	0.023	−0.121
α-Linolenic	−0.212	0.346
Arachidic	−0.024	−0.379
Gondoic	0.406 *	−0.070
Eicosadienoic	0.248	0.229
Dihomo-γ-Linolenic	−0.127	0.186
Arachidonic	−0.109	0.014
Eicosapentaenoic	0.074	−0.145
Behenic	0.304	−0.052
Adrenic	−0.182	−0.364
Docosapentaenoic n6	−0.392	−0.182
Docosapentaneoic n3	−0.209	0.050
Lignoceric	0.178	0.139
Docosahexaneoic	0.021	−0.007

Starred numbers indicate statistical significance. * *p <* 0.05.

**Table 5 cells-11-00910-t005:** Correlation between FA abundance and albumin stratified by cachexia status.

Fatty Acid	No Cachexia R	Cachexia R
Lauric	−0.533 **	0.006
Myristic	−0.010	0.010
Palmitic	−0.181	−0.225
Palmitoleic	0.049	−0.276
Stearic	−0.171	−0.058
Oleic	−0.099	−0.361
Vaccenic	−0.560 **	−0.646 **
Linoleic	0.116	0.577 *
γ-Linolenic	0.464 *	−0.432
α-Linolenic	0.367	0.014
Arachidic	−0.184	−0.293
Gondoic	−0.026	−0.025
Eicosadienoic	−0.056	−0.278
Dihomo-γ-Linolenic	0.098	−0.217
Arachidonic	0.167	−0.018
Eicosapentaenoic	0.102	−0.388
Behenic	0.021	0.175
Adrenic	0.221	−0.346
Docosapentaenoic n6	−0.008	−0.132
Docosapentaneoic n3	0.132	−0.134
Lignoceric	0.048	0.481
Docosahexaneoic	−0.090	−0.049

Starred numbers indicate statistical significance. * *p <* 0.05, ** *p <* 0.01.

## Data Availability

Data presented in this study are contained within the article.

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
