# Peer review of "Altered Plasma Fatty Acid Abundance Is Associated with Cachexia in Treatment-Naïve Pancreatic Cancer"

_cells, 2022, doi:10.3390/cells11050910_

Round 1

Reviewer 1 Report

it is a very well documented study 

Author Response

Thank you for your kind comment.

Reviewer 2 Report

The manuscript conducted small scale investigation through PDAC patient samples regarding cachexia and fatty acid content profiling. Overall the authors reported some interesting findings regarding some individual FA panels. The manuscript also justified why some omega 3 supplement would not help with cachexia in PDAC. I found the paper generally well prepared. Only some figure improvement is suggested.

All bar figures are presented in black and grey. It would be beneficial to turn into colored bars/dots like green/red. 

Author Response

We have changed our figures to only show individual values and used color-blind safe colors to enhance the legibility of the figures.

Reviewer 3 Report

while the work is very interesting, some concerns should be addressed carefully to solid the conclusions.

  1. In Table 1, the author needs to mark the statistical difference, and make a corresponding description in the table note.
  2. The clarity of the pictures and text in Figure 2 and Figure 3 is not enough, please replace the pictures with higher pixels.
  3. Some of the references in the manuscript are too old, and the authors are advised to replace the latest literature.

Author Response

  1. In Table 1, the author needs to mark the statistical difference, and make a corresponding description in the table note.

Response: We added a table footer indicating the bolded p-values in this table indicate significance.

  1. The clarity of the pictures and text in Figure 2 and Figure 3 is not enough, please replace the pictures with higher pixels.

Response: We have replaced the figures for better quality. Thank you for highlighting this for us.

  1. Some of the references in the manuscript are too old, and the authors are advised to replace the latest literature.

Response: We have looked through our reference list and found a few references where there are newer references that apply to our study. The two references from the 1950’s reflect the methodology we used for fatty acid isolation and measurement and we cannot change this.

Round 2

Reviewer 2 Report

The authors have addressed my suggestions.

Reviewer 3 Report

I support articles published on this edition